# Efficacy of Exercise on Muscle Function and Physical Performance in Older Adults with Sarcopenia: An Updated Systematic Review and Meta-Analysis

**DOI:** 10.3390/ijerph19138212

**Published:** 2022-07-05

**Authors:** Haolin Wang, Wendy Y. Huang, Yanan Zhao

**Affiliations:** 1School of Sports Science and Physical Education, Nanjing Normal University, Nanjing 210023, China; haolin_wang@njnu.edu.cn; 2Department of Sport, Physical Education and Health, Hong Kong Baptist University, Hong Kong SAR 00852, China; wendyhuang@hkbu.edu.hk

**Keywords:** exercise, older adult, sarcopenia, muscle strength, muscle mass, muscle function, physical performance

## Abstract

This study aimed to analyze the efficacy of exercise interventions on muscle strength, muscle mass, and physical performance in older adults with sarcopenia. Randomized controlled studies assessing exercise effects on sarcopenia were searched in Web of Science, PubMed, Cochrane Library, ProQuest, EBSCOhost, Scopus, EMBASE, and VIP and CNKI up to 31 March 2022. Data were expressed as weighted/standardized mean difference (MD/SMD) with 95% confidence intervals (CI). I^2^ index was employed for heterogeneity. The initial search identified 5379 studies, and 23 studies involving 1252 participants met the inclusion criteria for further analysis. Results revealed that exercise interventions can significantly improve grip strength (MD = 2.38, 95%CI = 1.33–3.43), knee extension strength (SMD = 0.50, 95%CI = 0.36–0.64), muscle mass of lower extremities (MD = 0.28, 95%CI = 0.01–0.56), walking speed (SMD = 0.88, 95%CI = 0.49–1.27), and functional mobility (MD = −1.77, 95%CI = −2.11–−1.42) among older adults with sarcopenia. No significant exercise effects were found on fat-free muscle mass, appendicular muscle mass, skeletal muscle mass, and muscle mass of the upper extremities. The results of subgroup analysis indicated that both resistance training and multicomponent exercise could significantly increase the muscle strength, while aerobic exercise did not. The findings suggest that exercise intervention can effectively improve muscle function and physical performance in older adults with sarcopenia, but has limited effects on the muscle mass of the upper extremities. In addition, it is highly recommended to apply group-based and supervised resistance training and multicomponent exercise in the prevention and treatment of sarcopenia among the older population.

## 1. Introduction

The remarkable increase in life expectancy and decrease in fertility rates result in an expanding ageing population worldwide. Because of ageing-related functional reduction, the wellbeing of older adults has been a focus for ageing-related research. A fact related to the advancing age is that older adults tend to have a sedentary lifestyle [1], which finally accelerates the loss of degradation of skeletal muscle mass and physical function.

Sarcopenia, as a critical component of frailty syndrome, refers to the progressive and generalized skeletal muscle disorder that involves the accelerated loss of muscle mass and function [2,3]. People with sarcopenia have been reported with declines in muscle strength and muscular functioning, which may ultimately lead to physical disability, reduced quality of life, and even death [4,5]. Although there are conflicting opinions about the diagnosis of sarcopenia, muscle strength, muscle mass, and physical functioning are the core diagnostic criteria for sarcopenia [6]. The accumulated evidence has indicated that sarcopenia is partially reversible, highlighting the importance of proper and early interventions for sarcopenia [7,8].

Among different interventions for sarcopenia, exercise has been well-evidenced to increase physical fitness, including but not limited to muscle mass, muscle strength/endurance, and cardiovascular capacities of older adults [7,9], highlighting its potential benefits to the prevention and treatment of sarcopenia. As a result of the literature review, most exercise interventions were conducted in healthy older adults or those at risk of sarcopenia [10,11,12]; while few in individuals with sarcopenia. Compared with those at the early stage of sarcopenia, people with sarcopenia would have a significantly lower physical function and thus show different responses to exercise interventions. The variations in participants would be one of the underlying reasons for the inconsistency in exercise efficacy for sarcopenia.

Even in individuals with sarcopenia, no consistent opinions have been reached regarding exercise efficacy. Lu and his colleagues found significantly improved walking speed after a 6-month exercise program in older adults with sarcopenia [13], while Zhu [14], Karina [15], and Iranzo [16] found limited improvements in walking speed, respectively. Inconsistent findings also exist in the related reviews. Vlietstra [17] found that exercise can significantly improve the muscle mass of older adults with sarcopenia, while Bao [18] did not found that. Therefore, this study systematically assessed the effects of exercise interventions on muscle strength, muscle mass, and physical performance in older adults with sarcopenia.

## 2. Materials and Methods

This study was conducted strictly following the reporting guidelines for systematic review and meta-analysis [19], and the study protocol was registered in the International Prospective Register of Systematic Reviews (CRD42021255735).

### 2.1. Search Strategy

We systematically searched the related randomized controlled trials (RCT) by implying exercise as the primary intervention among older adults with sarcopenia in the following databases, Web of Science, PubMed, Cochrane Library, ProQuest, EBSCOhost, Scopus, EMBASE, China National Knowledge Infrastructure (CNKI), and Chinese Science and Technology Periodical Database (VIP) from inception to 31 March 2022. The following MeSH terms were searched in English and Chinese: randomized controlled trial, sarcopenia, exercise, muscle function, physical performance, older adult, and the related terms used in their broadest sense. The detailed search strategies are described in Appendix A.

### 2.2. Eligibility Criteria

Inclusion criteria: (i) full-text published in English or Chinese; (ii) RCTs; (iii) taking exercise as the single intervention component in one group; (iv) participants were identified with sarcopenia; (v) participants aged 60 years and above; and (vi) primary outcomes include muscle function and physical performance indicators.

### 2.3. Study Selection and Data Collection

After removing the duplications, two researchers independently screened the study titles and abstracts. Then the full-text reviews were conducted with data extraction. Gray literature was explored by reviewing reference lists of selected primary studies and the related systematic reviews. Disagreements between the two researchers were resolved by consensus or a third researcher. The data included authors, publication year, sex, age, sample size, exercise interventions (types and intensity), and outcome measures. In cases of missing valuable data or information, the corresponding authors of these studies were contacted.

### 2.4. Risk of Bias Assessment

The methodological quality of the included studies was assessed using the Cochrane Collaboration risk of bias tool [20], which includes bias in selection (randomization and allocation), performance, detection, attrition, reporting, and other components. Two researchers performed the assessment independently, and the third researcher was consulted for resolution whenever discrepancies existed.

### 2.5. Statistical Analysis

The data analysis was performed using statistical software of the Cochrane Collaboration Review Manager (RevMan, version 5.3, Copenhagen, Denmark). Sample size, means, and standard deviations (SDs) were extracted for each RCT arm. The I^2^ statistic was used to describe the percentage of total variation across studies due to heterogeneity rather than chance [21]. A fixed-effect model was used to pool the results if the heterogeneity was not significant (I^2^ < 50%), and the random-effect model was applied when I^2^ > 50% [22]. Sensitivity analysis was carried out by removing studies one by one [23], and the subgroup analysis was performed for studies with high heterogeneity. Under the fix-model analysis, the weighted mean difference (MD) was used to assess the measure of effect, and the standardized mean difference (SMD) was applied if there were different outcome measures [24]. Finally, the STATA version 16 was used to assess the publication bias through the funnel plots and Egger test [25]. The statistical significance level was set at *p* < 0.05.

## 3. Results

### 3.1. Search Results

A total of 5379 studies were found in the initial search, from which 3335 duplications were removed, and 1948 records were removed for other reasons. Ninety-one studies were reviewed in full text, and 23 RCT studies involving 30 comparisons were included in this systematic review and meta-analysis (Figure 1).

### 3.2. Study Characteristics

The 23 RCT studies involving 1252 participants (665 received exercise interventions) were published between 2012 and 2022. Participants were aged from 60 to 101 years, with the mean age ranging between 63.2 ± 1.4 and 89.5 ± 4.4 years. Twelve studies included both men and women [14,16,26,27,28,29,30,31,32,33,34,35], while eight studies were conducted in women [15,36,37,38,39,40,41,42] and one study was conducted in older men [43]. Two studies did not specify the sex information [44,45]. The detailed information for data extraction can be found in Table 1.

Regarding the intervention content, different exercise types were used, including resistance training [15,26,28,31,34,35,39,41,42,43], aerobic exercise (e.g., Taichi, Qigong and Yi Jin Jing exercise) [26,28,29,32,33,34,43,45], and combined exercise [14,26,27,28,30,31,36,38,40,44]. The total duration of the exercise interventions ranged from 8 to 24 weeks, and most studies (*n* = 12) applied the 12-week duration [14,16,27,29,30,36,37,38,39,40,41,45]. Twice a week was the most adopted exercise frequency (*n* = 10) [14,15,26,27,28,31,36,37,38,44].

Variations exist in sample size among the included studies. Sample size ranged from less than 30 [15,40,41,42] to 80. Eleven studies included over 60 participants [14,26,28,29,30,31,36,37,38,43], and 10 studies comprise of 30–59 participants [16,27,30,32,33,34,35,39,44,45]. In addition, only half of the included studies reported detailed methods for sample size estimation [14,15,16,27,28,30,33,36,38,39,40,41].

### 3.3. Summary of Risk of Bias

For the selection bias, all studies reported how random sequences were generated, and 10 of 23 trials reported the allocation concealment [15,26,27,33,36,37,38,39,41,45]. Twenty-two studies did not conceal the research purpose to the participants, and one study [40] did not mention whether informed consent was signed or not. Therefore, there is a high risk of performance bias in these studies and this is the common problem in exercise interventions since blinding participants from exercise groups is quite difficult. Only five studies blinded the outcome assessment to participants [16,36,37,38,41], and 18 studies had a high risk of detection bias. In the two studies [16,26], a large number of subjects were lost in the experimental process, but the specific reasons were not reported, so there was a high risk of attrition bias. In addition, all studies had low risk regarding reporting bias and other biases. The detailed information can be found in Figure 2.

### 3.4. Effects of Exercise on Muscle Function and Physical Performance

Muscle function is the primary outcome of this study, and it includes muscle strength (i.e., grip strength and knee extension strength) and muscle mass (i.e., muscle mass of lower extremities, free fat mass, skeletal muscle mass, appendicular muscle mass, and muscle mass of upper extremities). The secondary outcome is physical performance, including walking speed and functional mobility as tested using the TUG.

#### 3.4.1. Grip Strength

A total of 17 studies involving 954 participants had taken grip strength as one of the main outcomes. A relatively high level of heterogeneity was present among different studies (I^2^ = 57.0%), and the sensitivity analysis indicated that removing any one study cannot lower the heterogeneity. The high heterogeneity may be due to different types of exercise formats. Resistance training was used in nine studies, aerobic training was applied in five studies, and multicomponent training was adopted in seven. Subgroup analysis was performed (Figure 3a), and the results demonstrated that exercise format is the main source of heterogeneity (I^2^ = 56%, *p* < 0.001). The intra-group heterogeneity was low in each of the three exercise formats (RT: I^2^ = 33%, AT: I^2^ = 34%; MT: I^2^ = 39%). The intervention effect size with the fixed-effects model demonstrated that resistance training (MD = 4.31, 95%CI = 3.22–5.39, *p* < 0.001) and multi-component exercise (MD = 1.59, 95%CI = 0.62–2.56, *p* = 0.001) can significantly improve grip strength of the target group. Aerobic exercise, however, had a limited effect on the improvement of grip strength (MD = 0.83, 95%CI = −0.58–2.24, *p* = 0.25). At the same time, we also adopted a random effect model combining the effect size of exercise interventions on grip strength (Figure 3b). The result demonstrated that exercise interventions can significantly improve the grip strength of the target group (MD = 2.38, 95%CI = 1.33–3.43, *p* < 0.001).

#### 3.4.2. Knee Extension Strength

Eight studies involving 620 participants applied knee extension strength as the outcome. Since the measures of knee extension were different (isometric measures vs. isokinetic measures), the SMD was applied to calculate the effect size, and a low level of heterogeneity was found (I^2^ = 12%). Results of the meta-analysis indicated that exercise intervention can significantly improve muscle strength for knee extension (SMD = 0.50, 95%CI = 0.36–0.64, *p* < 0.001) (Figure 4a).

In order to evaluate the effect of different exercise formats on knee extension strength in older adults with sarcopenia, a subgroup analysis was performed. Results showed that significant and positive effects of both resistance training (SMD = 0.84, 95%CI = 0.43–1.26, *p* < 0.001) and multi-component exercise (SMD = 0.54, 95%CI = 0.37–0.71, *p* < 0.001) on knee extension strength, while little is shown in the aerobic exercise on grip strength and is limited (SMD = 0.23, 95%CI = −0.06–0.51, *p* = 0.12) (Figure 4b).

#### 3.4.3. Muscle Mass

Eleven studies involving 648 participants set muscle mass as the main outcome, including muscle mass of lower extremities, free fat mass, skeletal muscle mass, appendicular muscle mass, and muscle mass of upper extremities. Results from sensitivity analysis indicated low heterogeneity among different studies (I^2^ = 0). Hence, the effect size was estimated using the fixed-effects model, and the results revealed significant exercise effects on the muscle mass of lower extremities (MD = 0.28, 95%CI = 0.01–0.56, *p* = 0.04) (Figure 5a), and limited effects were found in other muscle mass indicators of older adults with sarcopenia (Figure 5b–e).

#### 3.4.4. Walking Speed

There were 14 studies involving 872 participants that applied walking speed as the outcome. The study heterogeneity is quite big (I^2^ = 85%). Subgroup analysis of three different exercise types showed high heterogeneity within each exercise format (I^2^ > 80%). This would be related to the fact that big variations exist in test procedures and requirements. Among these studies, different walking distances were used including 4 m [27], 5 m [38], 6 m [14], 10 m [15,16,45], and 400 m [44]. Some studies also assessed walking speed by calculating walking distance in 10 min [39]. Inconsistencies also exist in starting position (static vs. dynamic) and whether or not to emphasize the ending point. The effect size of exercise intervention under the random-effects model indicated that exercise could significantly improve the walking speed of older adults with sarcopenia (SMD = 0.88, 95%CI = 0.49–1.27, *p* < 0.001) (Figure 6a).

#### 3.4.5. Functional Mobility

Eight studies involving 364 participants assessed functional mobility using the time up ang go test (TUG). The heterogeneity among these studies was low (I^2^ = 27%), and sensitivity analysis showed that the exclusion of any single study had no significant effect on the total effect size. Results from the meta-analysis using the fixed-effect model showed that, in comparison with the placebo group, exercise intervention can improve functional mobility (MD = −1.77, 95%CI = −2.11–−1.42, *p* < 0.001) (Figure 6b).

### 3.5. Publication Bias

The funnel plot of the knee extension strength was asymmetrical (Appendix A). The Egger tests also revealed that there was a relatively higher level of publication bias in knee extension strength (*t* = 2.11, *p* = 0.048). The funnel plots of other indicators were symmetrical, and Egger tests also indicated a relatively low level of publication bias (*p* > 0.05).

## 4. Discussion

This meta-analysis of 23 RCT studies assessed the effects of exercise interventions on muscle function and physical performance in older adults with sarcopenia. Results indicated that exercise could effectively improve muscle strength, muscle mass of lower extremities, and physical performance. In addition, resistance and multicomponent exercises would be preferred for muscle strength improvement. Results from this study would enrich the evidence of exercise efficacy in reducing the risks of sarcopenia among older adults.

Muscle strength is an often applied indicator in various diagnostic criteria of sarcopenia [46]. The present results found that moderate resistance training, multicomponent exercises such as aerobic + resistance training [14,26,28,40], resistance + balance + gait training [36,37], and resistance training + outdoor activity [30], would have significant effects on muscle strength for the target population. However, moderate aerobic exercise has shown a limited effect on muscle strength improvement. It has been well documented that resistance training can generate increases in muscle strength [47], whereas aerobic exercise is known to improve cardiorespiratory fitness [48]. In addition, subjective methods such as the self-perceived exertion scale were often used in aerobic exercise to monitor exercise intensity, which would influence the expected quality of aerobic exercise. Compared to multicomponent exercise formats, the single aerobic exercise training may not be interesting and attractive enough; therefore, it would reduce participants’ compliance. Regardless of the exercise formats, group exercise with supervision and guidance instead of individual home-based exercise always show better motivation for exercise involvement and more efficacy in muscle strength improvements [27]. It is highly recommended that clinical practice enriches the content of exercise, stimulates exercise interest, and adopts group-based supervised exercise intervention to strengthen the muscle function of older adults with sarcopenia.

The present finding confirmed the positive training effects on muscle mass, especially in the lower body, while Bao and his colleagues found no exercise effect on appendicular skeletal muscle mass [18]. The controversial results would be mainly related to differences in population characteristics. Compared with younger adults, older adults would show a slower physical response to resistance training [49] and gain a lower increment in muscle mass [50,51,52,53]. The muscle mass growth rate depends on the synthesis and degradation rate of proteins [54]. Due to the reduced level of physical activity and the retardation of muscle cell metabolism, the metabolism and synthesis of proteins are degraded among older adults. In addition, the presence of various chronic illnesses, including various inflammatory, respiratory, and endocrinal diseases, would reduce the synthesis rate of proteins in older adults [55]. As discussed above, older adults with sarcopenia have relatively lower levels of physical function and have limited tolerance to high exercise intensity. In addition, it has been well evidenced that exercise intensity significantly affects muscle protein synthesis [56]. All these would finally affect the enhancement of muscle mass amplitude among older adults with sarcopenia.

The present study revealed significant training effects on the muscle mass of lower extremities, which is in line with the findings of the previous related meta-analysis [17]. However, the present study revealed limited effects on the muscle mass of the upper body. This can be explained by the movement features of exercise intervention that tend to emphasize lower body muscle movement. In a study conducted by Kim and his colleagues (2012), the exercise program consisted of repeated toe raises, heel raises, knee lifts, knee extensions, lateral leg raises and leg extensions, hip flexions by resistance bands, but only arms pulling down and biceps bending for the upper body [36]. In addition to the standing posture during exercise intervention, the muscle mass of the lower extremities would receive more exercise stimulations compared to that of the upper body. Further RCT studies are recommended to include more upper body movements to exert substantial training effects on upper body muscle mass.

Walking speed is a commonly used indicator of physical functioning and is highly correlated with falling risks, cognitive ability, and neuromuscular function [57,58,59,60]. Both the Asian Working Group for Sarcopenia Standards (AWGS) and the 2nd Edition of the European Working Group on Sarcopenia Standards (EWGSOP2) include walking speed as the diagnosis criteria of sarcopenia. The results of this study confirm that exercise can effectively increase the walking speed of the target group, which is consistent with previous studies [61,62,63]. However, a relatively higher level of heterogeneity existed among the included studies. This would be mainly related to the differences in walking speed test parameters, such as starting position, walking distance, and the set of the ending points. All these would influence the outcome accuracy [64]. Due to ageing-related functional deterioration, older adults with sarcopenia would require a longer time in signal identification, muscle recruitment, and movement implementation than their age-matched counterparts. Future RCT studies are suggested to apply the standardized walking speed measurement to increase study comparison and reduce study heterogeneity.

There are some limitations in the present study that may comprise the application of study results. First, the quality of the included RCTs was not high, and most studies showed risks of performance bias and detection bias because of the improper application of blinding methods. This would influence the validity of the study results. Second, most studies were conducted on female participants; whether gender differences exist in training effects is unknown. Finally, given the limited number of the included studies, the present findings should be interpreted cautiously when translating into practice.

## 5. Conclusions

Exercise can effectively improve muscle function and physical performance in older adults with sarcopenia, but has limited effects on the muscle mass of the upper extremities. In addition, it is highly recommended to apply group-based and supervised resistance training and multicomponent exercise to prevent sarcopenia among the older population.

## Figures and Tables

**Figure 1 ijerph-19-08212-f001:**
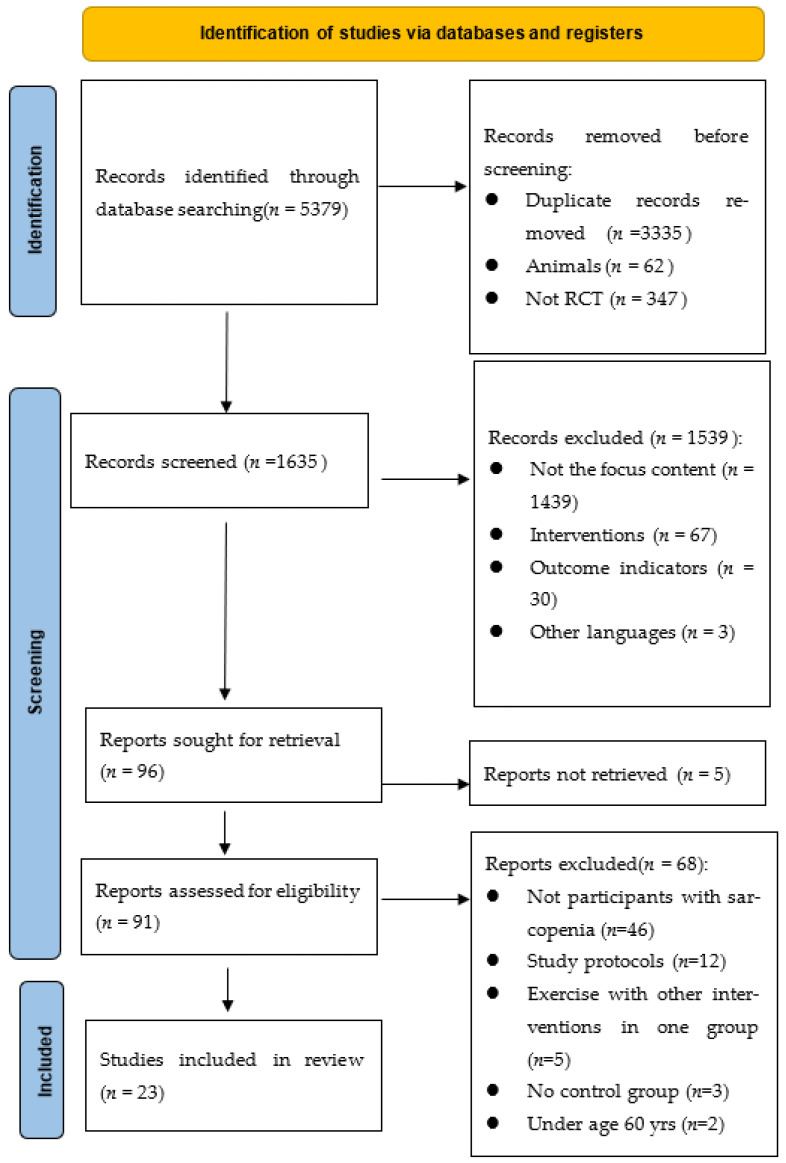
Flow diagram of the studies’ selection process.

**Figure 2 ijerph-19-08212-f002:**
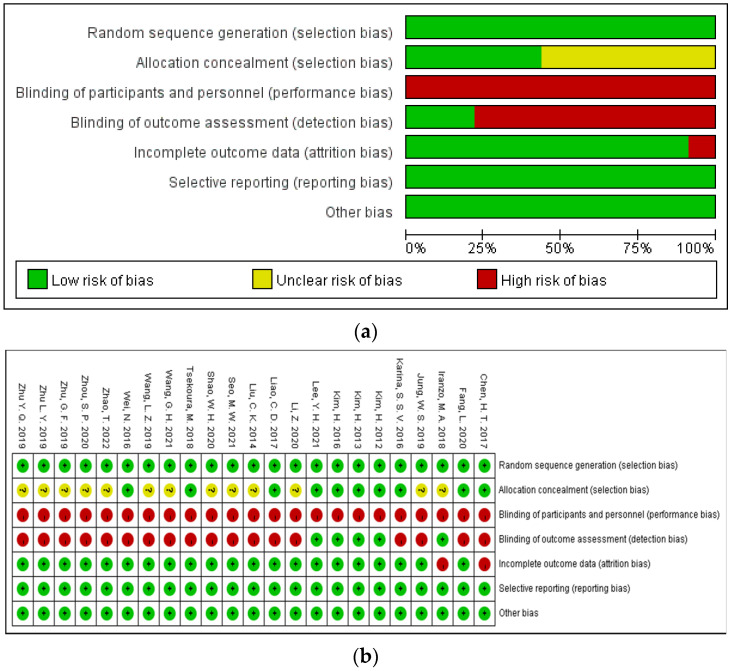
(**a**) The weighted plot for the assessment of the overall risk of bias; (**b**) summary of the bias for the trials included in this meta-analysis. Green indicates low risk of bias, yellow indicates unclear, and red indicates high risk of bias.

**Figure 3 ijerph-19-08212-f003:**
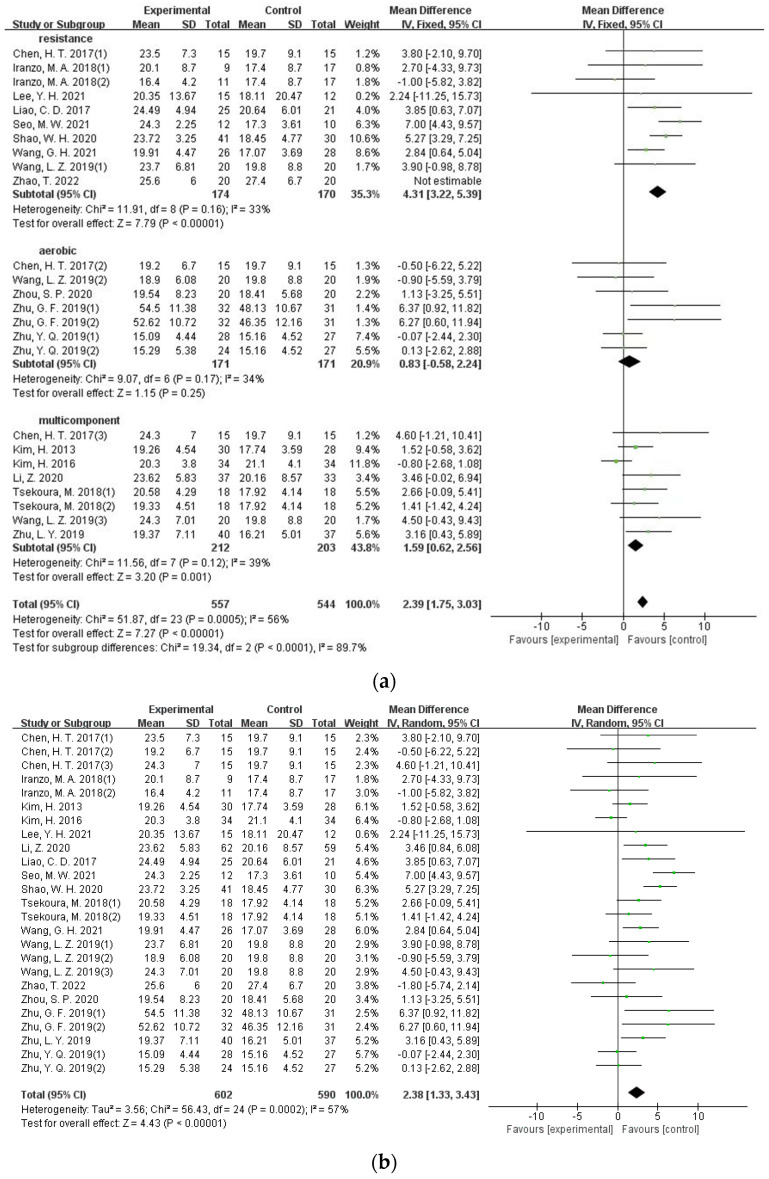
(**a**) Forest plot of the effectiveness of exercise interventions for grip strength improvement according to different exercise types; (**b**) forest plot of the effectiveness of exercise interventions for grip strength improvement in older adults with sarcopenia.

**Figure 4 ijerph-19-08212-f004:**
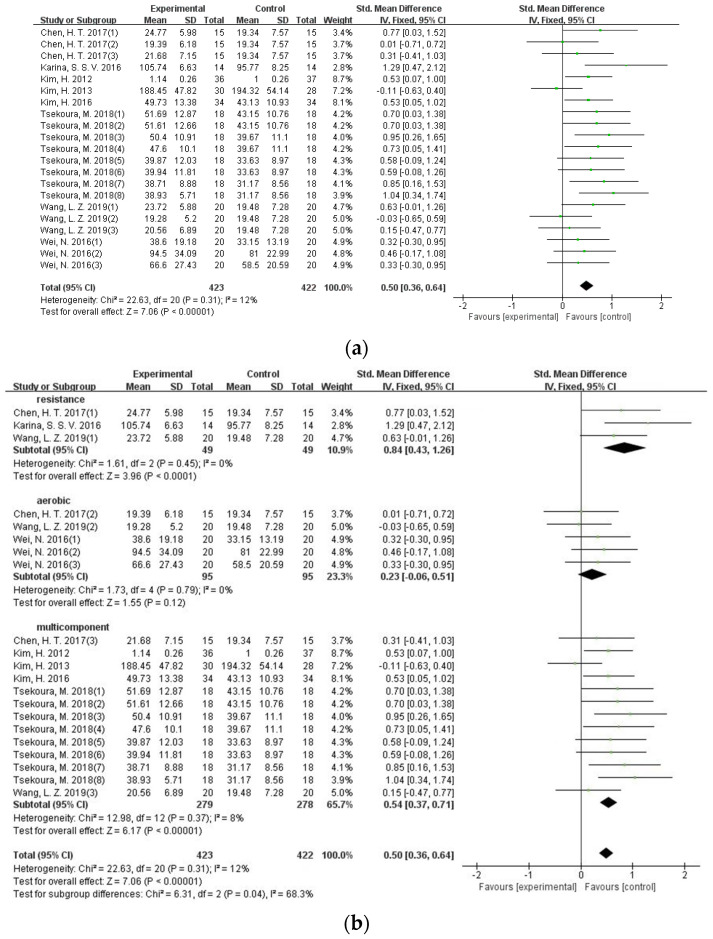
(**a**) Forest plot of the effectiveness of exercise interventions for knee extension strength improvement in older adults with sarcopenia; (**b**) forest plot of the effectiveness of exercise interventions for knee extension strength improvement according to different exercise types.

**Figure 5 ijerph-19-08212-f005:**
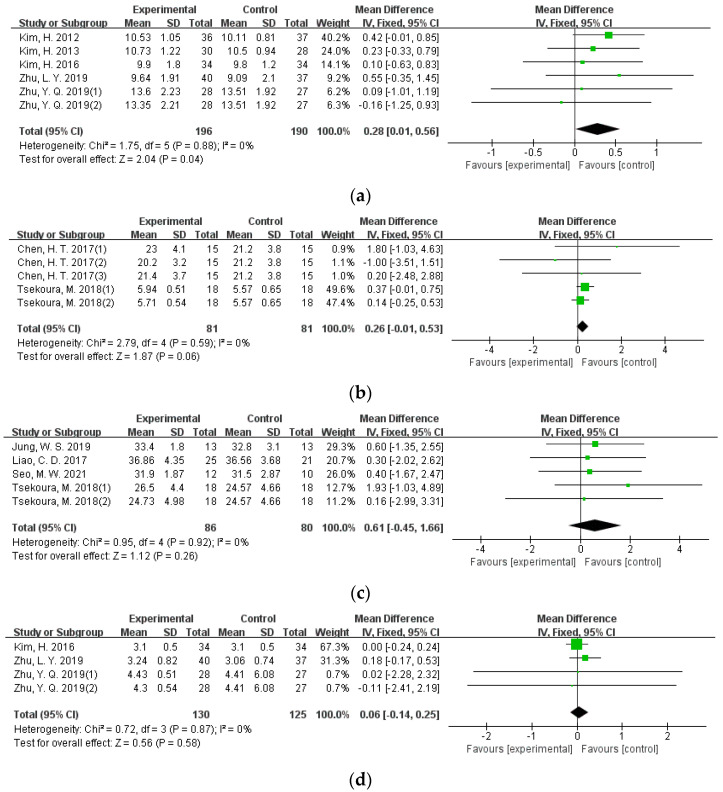
Forest plot of the effectiveness of exercise interventions for (**a**) muscle mass of lower extremities; (**b**) skeletal muscle mass; (**c**) free fat mass; (**d**) muscle mass of upper extremities; and (**e**) appendicular muscle mass improvement in older adults with sarcopenia.

**Figure 6 ijerph-19-08212-f006:**
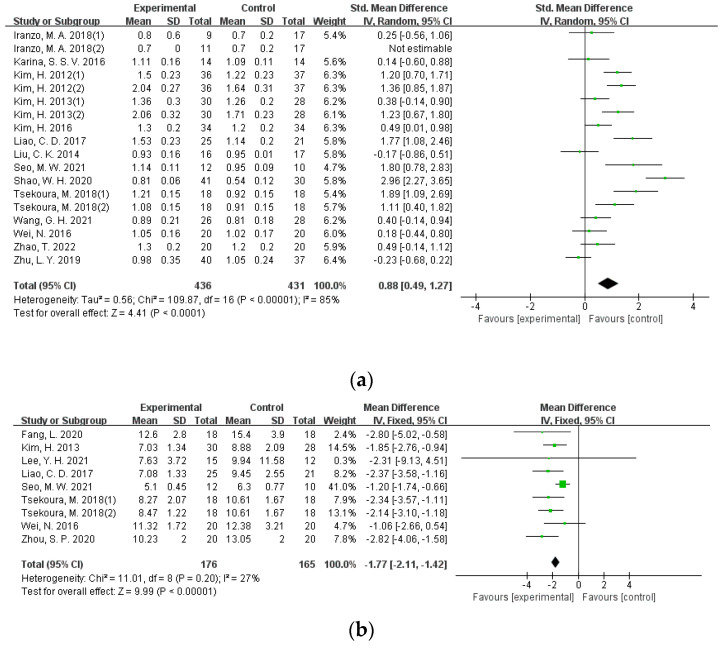
Forest plot for physical performance. (**a**) Forest plot of the effectiveness of exercise interventions for walking speed improvement in older adults with sarcopenia; (**b**) forest plot of the effectiveness of exercise interventions for functional mobility improvement in older adults with sarcopenia.

**Table 1 ijerph-19-08212-t001:** Study characteristics.

Author, Year, Country	Sample (Intervention/Control)	Age (Mean ± SD)	Sex	Interventions	Dosage	Intensity	Control Group	Outcomes
Zhu, L.Y. 2019, China [14]	77(40/37)	74.5 ± 7.1	Male (24%)	AT + RT	40–50 min × 2 times/week × 12 weeks (16–20 h)	Monitored and adjusted by coach.	Maintain daily lifestyle	Hand grip strength, muscle mass of lower extremities, muscle mass of upper extremities, walking speed
Karina, S.S.V. 2016, Brazil [15]	31(16/15)	72 ± 4.6	Female	RT	60 min × 2 times/week × 10 weeks (20 h)	Intensity between moderate and somewhat severe monitored by modified Borg Scale	Telephone monitoring	Knee extension strength, walking speed
Iranzo, M.A. 2018, Spain [16]	37(9,11/17)	82.6 ± 9.1	Male (25%)	RT	30 min × 3 times/week × 12 weeks (18 h)	Resistance:40–60% max isometric muscle strength	Maintain daily lifestyle	Hand grip strength, walking speed
Chen, H.T. 2017, China [26]	60(15,15,15/15)	68.9 ± 4.4	Male (16%)	RT/AT/AT + RT	60 min × 2 times/week × 8 weeks (16 h)	Resistance: 60–70% of the maximum repetitions aerobic: Moderate	Maintain daily lifestyle	Hand grip strength, knee extension strength, skeletal muscle mass
Tsekoura, M. 2018, Greece [27]	54(18,18/18)	72.87 ± 7	Male (13%)	AT + MT/AT + HT	AT + MT:40–50 min × 2 times/week × 12 weeks (16 h); AT + HT:30–35 min × 3 times/week × 12 weeks (18–21 h);	gradually increase: RPE from 6 to 20	Maintain daily lifestyle	Hand grip strength, knee extension strength, fat-free mass, skeletal muscle mass, walking speed, functional mobility
Wang, L.Z. 2019, China [28]	80(20,20,20/20)	65.1 ± 3.4	Male (53.8%)	RT/AT/AT + RT	20 min × 2 times/week × 8 weeks (5.4 h)	Moderate intensity: No obvious feeling of fatigue	Maintain daily lifestyle	Hand grip strength; knee extension strength
Zhu, G.F. 2019, China [29]	65(33/32)	66.32 ± 10.80	Male (50.1%)	Yi Jinjing exercise	40 min × 7 times/week × 12 weeks (56 h)	No obvious feelings of fatigue	Health education	Hand grip strength
Li, Z. 2020, China [30]	121(62/59)	73.73 ± 5.69	Male (37.1%)	RT + OA	20 min × 3 times/week × 12 weeks (18 h); 1 h × 3 times/week × 12 weeks (36 h)	RT:8 max rep.; OA: >800 steps in 10 min	Maintain daily lifestyle	Hand grip strength, appendicular muscle mass
Shao, W.H. 2020, China [31]	71(41/30)	69.3 ± 13.4	Male (63.4%)	RT	30 min × 2 times/week × 24 weeks (24 h)	Moderate intensity	Health education	Hand grip strength, walking speed
Zhou, S.P. 2020, China [32]	40(20/20)	73.0 ± 8.5	Male (42.5%)	Ba duanjin exercise	40 min × 5 times/week × 8 weeks (26.7 h)	Moderate intensity	Health education	Hand grip strength, functional mobility
Fang, L. 2020, China [33]	36(18/18)	82.8 ± 8.5	Male (33.3%)	Muscle-bone strengthening exercise	30 min × 3 times/week × 24 weeks (36 h)	60% maximum heart rate	Maintain daily lifestyle	Functional mobility
Wang, G.H. 2021, China [34]	54(26/28)	70.4 ± 5.1	Male (33.3%)	RT	45 min × 3 times/week × 12 weeks (27 h)	Moderate: 50-70% 1RM	Maintain daily lifestyle	Hand grip strength, walking speed
Zhao, T. 2022, China [35]	40(20/20)	63.2 ± 1.4	Male (40.0%)	RT	25 min × 14 times/week × 12 weeks (70 h)	8RM (can be adjusted according to individual situation)	Health education	Hand grip strength, walking speed
Kim, H. 2012, Japan [36]	78(39/39)	79.0 ± 2.9	Female	RT + Balance + Gait training	50 min × 2 times/week × 12 weeks (20 h)	Moderate: RPE = 12–14	Health education	Knee extension strength, muscle mass of lower extremities, appendicular muscle mass, walking speed
Kim, H. 2013, Japan [37]	64(32/32)	79.6 ± 4.2	Female	RT + Balance + Gait training	50 min × 2 times/week × 12 weeks (20 h)	Moderate: RPE = 12–14	Health education	Hand grip strength, knee extension strength, Muscle mass of lower extremities, appendicular muscle mass, walking speed, functional mobility
Kim, H. 2016, Japan [38]	69(35/34)	81.4 ± 4.3	Female	AT + RT and weight-bearing training	60 min × 2 times/week × 12 weeks (24 h)	AT: start at 40 watts and gradually increase; RT: beginning with 1 set of 10 repetitions to 3 sets.	Health education	Hand grip strength, knee extension strength, muscle mass of lower extremities, muscle mass of upper extremities, appendicular muscle mass, walking speed
Liao, C.D. 2017, China [39]	46(25/21)	67.3 ± 5.2	Female	RT	35–40 min × 3 times/week × 12 weeks (21–24 h)	Moderate: RPE = 13	No control	Hand grip strength, fat-free mass, functional mobility
Jung, W.S. 2019, Korea [40]	26(13/13)	75.0 ± 3.9	Female	AT + RT	25–55 min × 3 times/week × 12 weeks (15–60 h)	60–80% heart rate reserve	Maintain daily lifestyle	Fat-free muscle mass
Lee, Y.H. 2021, China [41]	27(15/12)	70.13 ±4.41	Female	RT	40 min × 3 times/week × 12 weeks (24 h)	Moderate intensity: RPE = 12–14	Allowed to exercise at home	Hand grip strength, functional mobility
Seo, M.Y. 2021, Korea [42]	27(14/13)	70.3 ± 5.38	Female	RT	50 min × 3 times/week × 16 weeks (32.5 h)	0—extremely easy to 10—extremely hard	Maintain daily lifestyle	Hand grip strength, functional mobility, walking speed, appendicular muscle mass, fat-free mass
Zhu, Y.Q. 2019, China [43]	79(24,28/27)	89.5 ± 4. 4	Male	Eight style TC/WBV	20 min × 5 times/week × 8 weeks (13.3 h)	TC: Progressively increase; WBV: 5 groups/time, and 3 min/group	Maintain daily lifestyle	Hand grip strength, muscle mass of lower extremities, muscle mass of upper extremities
Liu, C. K. 2014, America [44]	33(16/17)	77.5 ± 4.2	No clear	AT + RT + Balance + Flexibility training	1–8 week:3 times/week; 9–24 week:2 times/week	Moderate	Not clear	Walking speed
Wei, N. 2016, China [45]	40(20/20)	75 ± 6	No clear	WBV	6 min × 3 times/week × 12 weeks (3.6 h)	40 Hz/360 s per session;	Maintain daily lifestyle	Knee extension strength, walking speed, functional mobility

AT, aerobic training; RT, resistance training; MT, multicomponent training; OA, outdoor activity; HT, home therapeutic exercise; TC, Tai Chi; WBV, whole-body vibration training; RPE, rating of perceived exertion.

## Data Availability

Not applicable.

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
