# Peer review of "Efficacy of Exercise on Muscle Function and Physical Performance in Older Adults with Sarcopenia: An Updated Systematic Review and Meta-Analysis"

_ijerph, 2022, doi:10.3390/ijerph19138212_

Round 1
Reviewer 1 Report
The topic of this manuscript falls within the scope of IJERPH Journal. The topic of the manuscript is very interesting and original.
This study aimed to analyze the efficacy of exercise interventions on muscle strength, muscle mass, and physical performance in older adults with sarcopenia.
The Authors showed that exercise can effectively improve muscle function and physical performance in older adults with sarcopenia, but has limited effects on the muscle mass of the upper extremities. They also recommend applying group-based and supervised resistance training and multicomponent exercise in the prevention and treatment of sarcopenia among the older population
The Authors have presented sufficient data. The appropriate tables and figures have been provided. The article is easy to read and logically structured. The methods are adequately described. The authors used appropriate statistical methods. The conclusions are consistent with the presented evidence and arguments.
the strength of this paper: very interesting topic; material and methods-the right choice of methodology methods, which was presented incomprehensible way; the obtained results are presented in the form of figures and tables, which are clear and easy to understand; the discussion- supports the results properly and refers to the current literature in an appropriate manner; the conclusions- based on the obtained results, they are consistent with evidence and arguments. They address the main question posed.
There are some comments in the reviewer's opinion that should be taken under consideration by the Authors:
1. Please add the limitation of your work
2. Please discuss that sarcopenia is considered to be a key component of frailty syndrome
[please cite: doi: 10.3390/ijerph17249339; doi: 10.1016/j.cger.2017.09.004]
Reviewer 2 Report
Dear Authors,
The meta analysis is well designed and about an important parameter of sarcopenia treatment ''excercise''.
There are other several methods for sarcopenia treatment as ''vitamin D'' and ''protein''. This data may be needed in review articles not in meta analysis. Thus the title should be ''Efficacy of exercise on muscle function and physical performance in older adults with sarcopenia: an updated meta-analysis''.
Best regards
Reviewer 3 Report
This review paper is based on well analyzed more than five thousand materials where effect of exercise was studied on muscle function (muscle mass, strength, walking speed...) in older adults suffering with sarcopenia. Resistant exercise effect on sarcopenic elderly muscle function was significant, but not aerobic type of exercise. There was no effect of exercise on sarcopenic muscle fat free muscle mass , and limited effect on upper extremities. Resistant exercise have the preventive roll of the of sarcopenia in the elderly population.
At first glance it seems me that lack of paper is absence of description on cellular and molecular level, but this not necessary as the aim of paper is another.
Well written review article.
Reviewer 4 Report
Introduction
36-40 Rephrase
42 Definition not correct also reference is not the most appropriate
43 Probably Authors need to explain what is the relationship between Sarcopenia and Frailty (see Ferrucci or alternatively Guralnick or Fied)
45 This is the correct definition, even if a little caution needs to be applied considering in the definition physical punction (discuss)
47 Tremendous what AA need “important” “pivotal” not clear
48 Rephrase repetitive
56 Fitness is somehow a different concept from Function, in the sarcopenic patients is more accurate function.
60 This is not true, exercise is the only “therapeutic” approach to prevent and revert sarcopenia, probably we can discuss about type of exercise that could the best approach.
62-70 I found very strange that from this review-metanalysis, only (mostly) Chinese study were considered, probably a larger assessment of international bibliography could help. Moreover, mostly are in Chinese languages and is difficult to consider. This paper as it is, could be more interesting or understood from a specific geographic area.
Methods
72 Reference
74 Number
76-82 Is difficult at least for this Referee to understand Chinese keywords (translation could be an option)
99 psychological is a typo?
117 Figure. (remove .)
129 3366 (97%) papers were excluded, without a clear reason? (They are conduct in a different country from China? Is this the reason?)
137 2 studies excluded for protocol could explain
157-158 Those approaches to exercise are clearly not adequate to sarcopenia in the elderly in terms of frequency and length of the intervention
159 Sample size determination need to be largely discussed (please refer to Pahor Marco to expand and discuss Sample size determination in elderly during exercise)
Summary of Risk Bias
I found very strange that attrition and reporting biases are low in your analysis
Effect of exercise (3.4)
Muscle strength definition, in this case I have some concern about the two types of muscles assessment; grip strength, probably the most used tools, but for knee extension strength the problem could be related to the type of muscles evaluation or to the type of instrument used (protocol). I think that this could be a problem of heterogeneity, so different protocol could not enable the authors to disentangle and compare the results.
Another concern deals with BMI included in physical performance? BMI is usual included in the models as confounder, and it is not considered as an outcome.
194 typo (?) 20 studies including 27 RCT.
212 similar problem, what authors mean with RCT (randomized clinical trials?) never referenced in the text, 8 studies involving 21 RCT.
Figure 3 A-E are not intelligible. Please change the scale.
Please pay attention Sarcopenia is not synonymous of muscle mass.
Figure 4 same problem of figure 3
Gait agility, never described in the methods, and what is the relationship between agility and Sarcopenia (?)
Figure 5 see figure 3-4
Discussion
278 Again Muscle mass or strength alone are not synonymous of Sarcopenia, what you have done is an analysis on strength or mass or performance, and you can only said that exercise improves those indirect markers of Sarcopenia.
Round 2
Reviewer 4 Report
36 Because of ageing-related degradations, wellness is essential to older adults and has been the focus in ageing-related research. This sentence is not derived from a clinical approach, is more reliable for sport therapist. I try to explain, what do you mean with “degradation” (functional reduction? Or clinical impairment, or disability), moreover wellness is a not a focus for age related researcher, instead could be: wellbeing, quality of life and/or independency. Reconsider de novo, please.
60-65 Is not clear what authors would like to say. Probably could be a problem of English construction? Try to be clearer, please.
61while Zhu [14] did not observe any changes in walking speed in their studies. There is only one reference, if are many studies please cite.
67 “this study included Randomized controlled trial (RCT) studies published in Chinese, which may provide a more comprehensive evaluation of exercise efficacy.” Why? Please give a more consistent reason. If this is the reason, I think that, this article is more appropriate for a Chinese Journal. The key point of meta-analysis is the comparison also with transcultural population.
Gait agility, is not a proper definition for performance test as time up and go, or SPPB. Those are performance test. Standing up from a chair is a measure of strength, is not intended as a proxy for agility.
Please, could you split figures, and explain a little better what is presented in the legend? Are very difficult to read.
Limitations, in this section You have to report that You have selected only Chinese Study, this solution obviously limit the external validity of your findings. Or better probably what you found is specific for Chinese population.
